# Role of Altitude in Formation of Diatom Diversity of High Mountain Protected Glacier Lakes in the Kaçkar Mountains National Park, Rize, Turkey

**Bülent Şahin** [1] and **Sophia Barınova** [2,*]

1  Department of Biology Education, Fatih Education Faculty, Trabzon University, 61335 Trabzon, Turkey
2  Institute of Evolution, University of Haifa, Mount Carmel, 199 Abba Khoushi Avenue, Haifa 3498838, Israel
*  Correspondence: sophia@evo.haifa.ac.il; Tel.: +972-4824-9799

**Abstract:** The benthic diatom assemblages of the glacier Avusor Great Lake and Koçdüzü Great Lake were investigated in August 2019. A total of 71 diatom species belonging to 34 genera were determined in the samples, 49 species from Avusor Great Lake and 37 from Koçdüzü Great Lake. Only 15 diatom species were common to both lakes. Total floristic similarity between the neighbouring lakes was only 21.12%. Genera with the highest number of species were *Eunotia* (8), *Gomphonema* (6), *Pinnularia* (6), *Navicula* (5) and *Aulacoseira* (4). The diatom flora of both lakes was formed by cosmopolitan species with a large influence from species in alpine and subalpine area. A comparison of the diatom assemblages of the investigated lakes showed differences in both relative abundance and species present in the individual lakes. Bioindicative analysis showed that the water of Koçdüzü Great Lake is more alkaline and less saturated with organic substances than Avusor Great Lake. In both lakes, the communities are composed of species adapted to living in the benthos of fresh waters of moderate temperature well enriched with oxygen. In both lakes, the water quality was Class 2 with a saprobity index of S = 1.08 in Avusor Great Lake and 0.97 in Koçdüzü Great Lake, but their communities were composed of species with both oligotrophic and mesotrophic status. Statistical comparison of the species composition of studied lakes with other high mountain lakes and the lakes in northern Turkey revealed the role of altitude as the main factor in the formation of diatom floras.

**Keywords:** high mountain; diatoms; benthic habitats; bioindicators; comparative statistics; Avusor Great Lake; Koçdüzü Great Lake; Turkey

## 1. Introduction

Mountainous regions have areas called "biodiversity hotspots" and these areas are home to many endemic and endangered species. Mountain ecosystems, which are an important source of water, energy, and biodiversity, are essential for the continuation of ecosystems. A significant percentage of the world's population benefits from resources in mountain ecosystems, primarily water. However, mountain ecosystems, which are highly vulnerable to human and natural ecological imbalances, are also very susceptible to climate change [1,2].

High mountain lakes have extreme environmental conditions such as low temperatures, low nutrients, short growing times, and high radiation, and they spend a large part of the year under ice and snow [3–5]. Therefore, they usually have an oligotrophic character [6]. Organisms that adapt to these extreme climatic and physico-chemical conditions create a biodiversity unique to high mountain lakes [3]. Examination of high mountain lakes, which are regarded as indicators of environmental degradation and global changes, is of great importance for the future [5,7–9].

Benthic algae are capable of photosynthesis. Therefore, they are among the oxygen and energy sources of aquatic ecosystems. They are also used as bioindicators since they react to environmental changes [10,11]. Diatoms are important organisms used in

determining water quality and monitoring changes in water quality [12]. Diatoms provide great advantages in the ecological monitoring of aquatic ecosystems because of the diversity of their populations, their easy sampling, they contain indicator species, and the structure of the communities being related to the ecological structure of water [13]. Therefore, the European Union has defined benthic diatoms as one of the organisms used to determine the ecological quality of water resources in the Water Framework Directive report, which it declared in 2000 [14].

The Eastern Black Sea Region is one of the ecosystems with the highest biological diversity in the world [15–19]. Rize is one of the most striking provinces of the region in terms of biodiversity due to its current climate characteristics, which has cool summers, mild winters, and rainy seasons during the whole year [20]. The data obtained in this study, which aims to determine the floristic and ecological characteristics of benthic diatom communities of Avusor and Koçdüzü Great Lakes, is the first record for the province of Rize and the studied lakes.

Algae as a whole, and diatoms, as one of the smallest and most diverse organisms in a lake ecosystem, represent the first trophic level of the trophic pyramid. They create proteins through photosynthesis and thus form the food base for other organisms that live in the lakes. Their absence or overgrowth can harm water resources and diversity. Several factors have been noted that can influence the growth and diversity of algae. We hypothesize that lake altitude will affect diatom diversity due to changes in ionic content and water temperature that influences oxygen concentration. Analysis of the composition of indicator species of oxygen saturation, salinity, and organic pollution will help to reveal the effect of altitude on the state of the lakes' ecosystem.

## 2. Materials and Methods

### 2.1. Study Area

Both studied glacial lakes are located within the borders of Çamlıhemşin district of Rize province on the territory of the Kaçkar Mountains National Park. Located at 40°56′11″ N–41°12′01″ E coordinates, the Avusor Great Lake has an altitude of 2678 m above sea level (a.s.l.), and a surface area of 2.2422 ha. The altitude of the Koçdüzü Great Lake is 2382 m a.s.l., and its surface area is 8.1896 ha. The lake is located at 41°00′15″ N–41°11′53″ E coordinates (Figures 1 and 2).

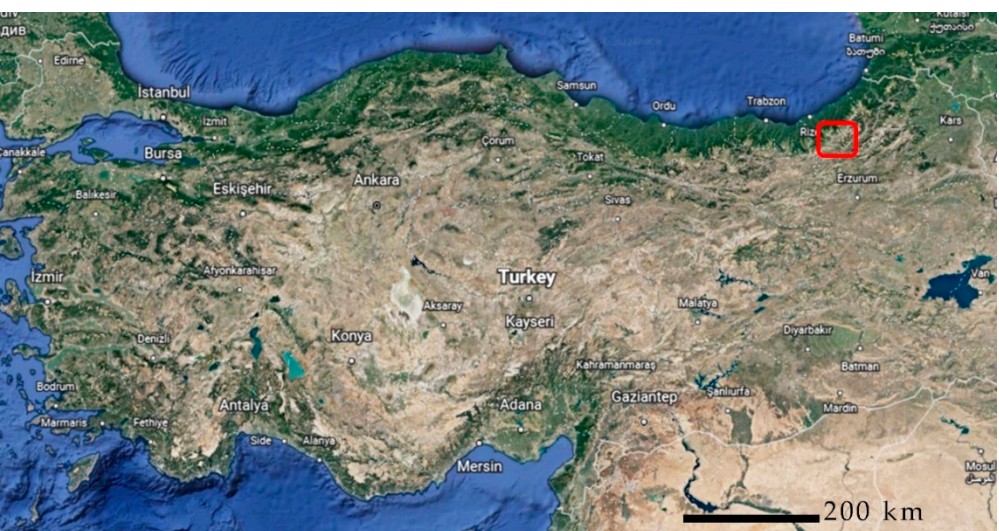

**Figure 1.** The location of the study area, red square.

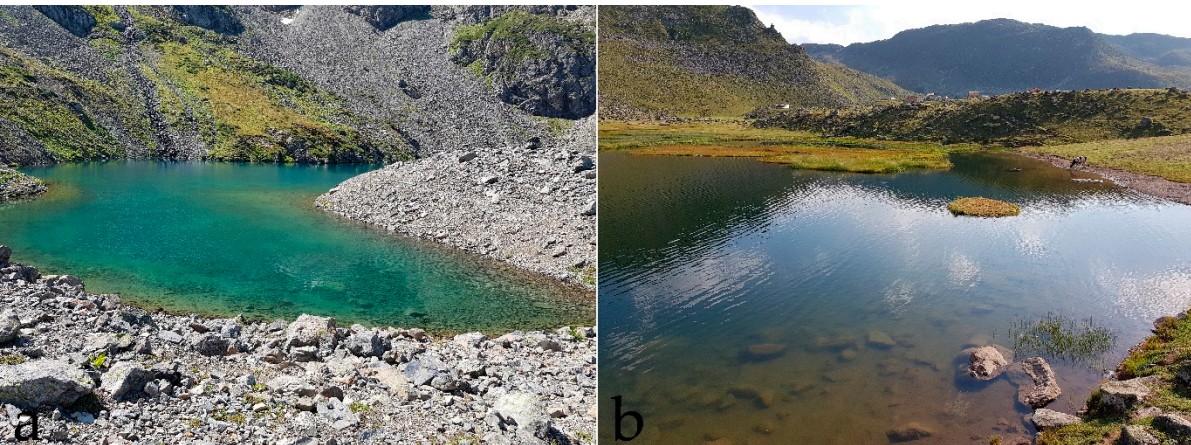

**Figure 2.** View of the Avusor Great Lake (AGL) (**a**) and Koçdüzü Great Lake (KGL) (**b**).

*2.2. Sampling and Laboratory Studies*

The diatom samples were collected from Avusor Great Lake and Koçdüzü Great Lake on 21 August 2019. Epipelic diatom samples were obtained with a glass tube, which is 1 m long and 0.8 cm diameter from the surface of the sediments of both lakes. Epilithic samples were obtained from only Avusor Great Lake, scraped from randomly chosen stones with the toothbrush, and washed into plastic bottles. Epiphytic diatoms were collected by squeezing out the macrophytes (*Potamogeton* sp. and *Juncus* sp.) from only Koçdüzü Great Lake [21,22]. All samples were fixed in 4% formaldehyde in the 100 mL plastic bottles. At the same time, water temperature, dissolved oxygen, conductivity, and pH were measured using Thermo Orion-4-Star pH (Hampton, NH, USA, Marshall Scientific) and YSI-55 (Letchworth, Hertfordshire, United Kingdom, Xylem Analytics) portable meters. In the laboratory, the slides were prepared to the method of Round [21] and mounted in Naphrax mounting media. Analyses of other hydrochemical parameters were carried out in the DSI General Directorate Laboratories DSI 22nd Regional Directorate Quality Control and Laboratory Branch Office. Diatoms were examined with the Leica DM 2500 light microscope and photographed with the Leica MC170 HD camera (Wetzlar, Germany, Leica Microsystems). Identification of the diatom species was made using the relevant handbooks [23–31]. The current scientific names of the species were updated according to algaebase.org [32]. The abundance scores estimation was made according to a 6-point scale [33]: 1—"single" with 1–5 cells per slide, 2—"rare" with 10–15 cells, 3—"common" with 25–30 cells, 4—"frequent" with one cell over a slide transect, 5—"very frequent" several cells over a slide transect, 6—"abundant" with one or more cells in each field of view.

The first experience of constructing distribution curves for the number of genera species by the number of algae genera was carried out earlier by J.C. Willis in the book "Age and Area" in 1922 [34] for different lists of faunas and floras of angiosperms, shows that the Willis curves constructed for the algal floras of various water bodies of Eurasia have a hyperbolic distribution only with a well-studied flora, and therefore the curve can be used as a criterion for the completeness of the list of algae [35].

The ecological preferences of the identified species were determined using bioindication methods [36]. For each species, its indicator properties were determined in relation to one or more environmental variables [37,38]. Then the data on the number of species in each indicator group were summarized. The distribution of the number of species with the same indicator properties was constructed with respect to each environmental variable. The class of water quality indicators were grouped on the range of species–specific index saprobity S: Class 1, S = 0.0–0.5; Class 2, S = 0.5–1.5; Class 3, S = 1.5–2.5; and Class 4, S = 2.5–3.5. Altogether, nine environmental variable indicator groups were used for analysis. The location of the indicator groups of each environmental variable on the histogram was in order of strength for this variable.

Bray–Curtis similarity analysis was carried out and a tree was constructed with the help of the BioDiversity Pro 9.0 program. Pearson coefficients of similarity were calculated in [39]. The correlation analysis of species content was conducted as the network plot in JASP (Jeffrey's Amazing Statistics Program 0.16.4) on the botnet package of Statistics with R [40]. Three-dimensional surface plots for the dependencies of individual parameters were built in the program Statistica 12.0 according to the Distance Weighted Least Square method. Three-dimensional surface plots were constructed based on the data for each of the parameters shown in the table for major variable distribution. For comparison, one main parameter and two others were selected for each graph, within the limits of which the program calculates probable changes in the main parameter. Thus, the resulting graph shows the trends in each of the related parameters. From here, outrageous values may appear, which are not real, but only reflect trends for a given distribution. The graph can be interpreted as a trend of change (increase or decrease) in the values of the main parameter (on the x axis) when the other two parameters (y and z axes) change.

## 3. Results

### 3.1. Physical and Chemical Analysis

The pH level was neutral in Avusor Great Lake, while it was alkaline in Koçdüzü Great Lake. The conductivity value was lower than 50 μS/cm$^{-1}$ in the Avusor Great Lake, while it was 104.7 μS cm$^{-1}$ in the Koçdüzü Great Lake. Avusor Great Lake had the highest concentration of total dissolved matter. While nitrate nitrogen was found in Avusor Great Lake, it was less than in Koçdüzü Great Lake. Magnesium was not found in either lake. The concentrations of ammonium nitrogen and nitrite nitrogen were almost the same in both lakes. The concentrations of phosphate in all lakes were low (Table 1).

**Table 1.** Averaged physical and chemical data, GIS coordinates and altitude of the Avusor Great Lake and Koçdüzü Great Lake.

| Variable | Avusor Great Lake | Koçdüzü Great Lake |
|---|---|---|
| North | 40°56′11″ | 41°00′15″ |
| East | 41°12′01″ | 41°11′53″ |
| Altitude | 2678 | 2382 |
| Temperature (°C) | 15.9 | 21.0 |
| Dissolved oxygen (mg L$^{-1}$) | 10.2 | 9.2 |
| pH | 7.58 | 8.45 |
| Conductivity (μS cm$^{-1}$) | 45.3 | 104.7 |
| Total Dissolved Matter (mg L$^{-1}$) | 28.01 | 6.88 |
| Potassium (mg/L) | 0.18 | 0.23 |
| Total Hardness CaCO$_3$ (mg L$^{-1}$) | 18.96 | 12.94 |
| Calcium (mg L$^{-1}$) | 5.53 | 3.12 |
| Magnesium (mg L$^{-1}$) | - | - |
| Ammonium (mg L$^{-1}$) | 0.14 | 0.16 |
| Chloride (mg L$^{-1}$) | 6.31 | 10.78 |
| Nitrate (mg L$^{-1}$) | 1.092 | - |
| Nitrite (mg L$^{-1}$) | 0.083 | 0.081 |
| Ammonium nitrogen (mg L$^{-1}$) | 0.11 | 0.12 |
| Nitrate nitrogen (mg L$^{-1}$) | 0.247 | - |
| Nitrite nitrogen (mg L$^{-1}$) | 0.025 | 0.024 |
| Medium phosphate (mg L$^{-1}$) | 0.07 | 0.072 |
| Phosphate (P$_2$O$_5$) (mg L$^{-1}$) | 0.053 | 0.054 |

Note: (-): Could not be detected as below the determination level.

### 3.2. Diatom Assemblages

A total of 71 species belonging to 34 genera were determined in the samples: 49 species from Avusor Great Lake and 35 from Koçdüzü Great Lake. In the first step of floristic analysis, the Willis curve was constructed for the revealed species composition in the studied lakes. Figure 3 shows that distribution of species number over number of genera in the diatom flora of the Avusor Great Lake (AGL) and Koçdüzü Great Lake (KGL) formed the line that was close to the trend line for the distribution with R = 0.89. Therefore, the

studied species list can be analyzed as the flora in respect of the taxonomic presentation [35]. It has been previously found [35,41] that the Willis curve can be a criterion for the fullness of a species list in well-studied algal floras in Eurasia. Therefore, we can carry out floristic, taxonomic, and ecological analysis for the diatom flora in AGL and KGL and compare it to other high mountain lake floras in this region where the Willis proportion also closely follows the hyperbolic shape [8,42–46].

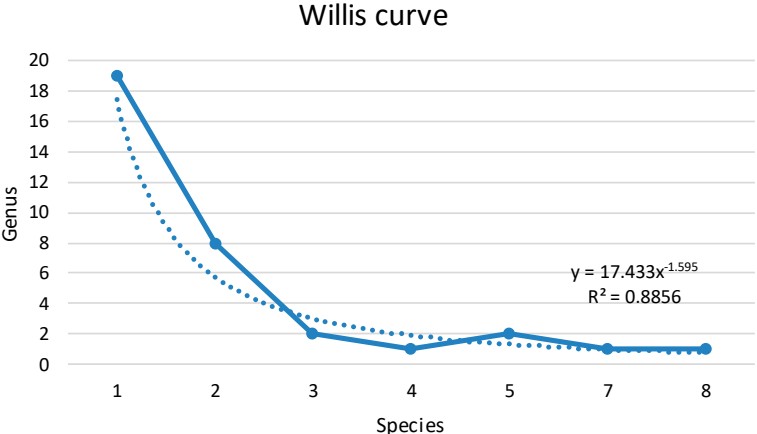

**Figure 3.** The Willis curve: distribution of genera number over species number in the diatom flora of the Avusor Great Lake (AGL) and Koçdüzü Great Lake (KGL). Dotted line is the trend line with $R^2$ = 0.8856.

When the diatom communities of the investigated lakes were compared, differences were observed in species composition and relative abundance of each lake. Abundance as a sum of species abundance scores was larger in Avusor Great Lake (97) than in the Koçdüzü Great Lake (69) that demonstrated more comfortable environment for diatom assemblage in AGL. As can be seen in Table 2, diatom species preferred the epipelic habitats more than epilithic or epiphytic in both lakes.

**Table 2.** List of identified diatoms in the Avusor Great Lake (AGL) and Koçdüzü Great Lake (KGL) with abundance scores and ecological preferences.

| Taxa | Epipelic | Epilithic | Epiphytic | AGL | KGL | Hab | Oxy | pH | pH Range | Sal | T | Tro | Aut-Het | D | Sap | S |
|---|---|---|---|---|---|---|---|---|---|---|---|---|---|---|---|---|
| *Achnanthidium minutissimum* (Kützing) Czarnecki | 1 | | | 2 | 0 | P-B | st-str | ind | 4.3–9.2 | i | eterm | e | ate | es | b | 0.95 |
| *Amphora ovalis* (Kützing) Kützing | 1 | | | 2 | 2 | B | st-str | alf | 6.2–9.0 | i | temp | e | ate | sx | b | 1.50 |
| *Aulacoseira italica* (Ehrenberg) Simonsen | | 1 | | 2 | 0 | P-B | st-str | ind | 5.8–8.4 | i | cool | me | ate | es | b | 1.45 |
| *Aulacoseira lacustris* (Grunow) Krammer | 1 | | | 0 | 2 | P | - | acf | - | hb | cool | ot | - | - | x-o | 0.50 |
| *Aulacoseira valida* (Grunow) Krammer | 1 | | | 0 | 2 | P-B | - | alf | - | i | - | om | ate | es | o | 1.30 |
| *Aulacoseira* sp. | 1 | | | 0 | 2 | - | - | - | - | - | - | - | - | - | - | - |
| *Brachysira brebissonii* R.Ross | 1 | | | 0 | 2 | P-B | st-str | acf | 4.6–7.8 | hb | temp | ot | ats | sx | o | 0.40 |
| *Caloneis alpestris* (Grunow) Cleve | 1 | | | 2 | 0 | B | str | alf | - | i | - | m | ats | - | o | 0.10 |
| *Caloneis silicula* (Ehrenberg) Cleve | 1 | | | 2 | 0 | B | st | ind | 6.3–9.0 | i | warm | om | ats | sp | o | 1.30 |
| *Cocconeis placentula* var. *euglypta* (Ehrenberg) Grunow | 1 | | | 2 | 0 | P-B | st-str | alf | 5.5–9.0 | i | temp | om | ate | sx | b | 1.30 |
| *Cymbella affinis* Kützing | 1 | | | 2 | 0 | B | st-str | alf | 6.3–9.5 | i | temp | e | ats | sx | b | 1.10 |
| *Cymbella cistula* (Ehrenberg) O.Kirchner | 1 | 1 | | 2 | 0 | B | st-str | alf | 8.0 | i | - | e | ats | sx | b | 1.20 |
| *Cymbella* sp. | | 1 | | 2 | 0 | - | - | - | - | - | - | - | - | - | - | - |
| *Cymbopleura amphicephala* (Nägeli ex Kützing) Krammer | 1 | | 1 | 0 | 2 | B | st-str | ind | 4.6–8.10 | i | - | om | ats | sx | o | 1.20 |
| *Diatoma vulgaris* Bory | 1 | | | 2 | 2 | P-B | st-str | alf | 6.2–8.9 | i | temp | me | ate | sx | b | 2.20 |
| *Diatoma* sp. | 1 | | | 2 | 0 | - | - | - | - | - | - | - | - | - | - | - |
| *Didymosphenia geminata* (Lyngbye) Mart.Schmidt | 1 | 1 | | 2 | 0 | B | st-str | ind | - | i | - | ot | ate | sx | o-x | 0.70 |
| *Diploneis elliptica* (Kützing) Cleve | 1 | | | 2 | 0 | B | str | alf | 8.2 | i | temp | m | ats | sx | o-x | 0.60 |
| *Encyonema minutum* (Hilse) D.G.Mann | 1 | 1 | | 1 | 0 | B | st-str | ind | 4.9–8.9 | i | temp | m | ate | sx | o | 1.20 |
| *Encyonema silesiacum* (Bleisch) D.G.Mann | 1 | | | 2 | 0 | B | st-str | ind | 6.2–8.6 | i | temp | o-e | ate | sx | o | 1.20 |
| *Epithemia gibba* (Ehrenberg) Kützing | 1 | | | 0 | 2 | P-B | st-str | alf | 6.2–9.0 | i | temp | om | ate | es | x-o | 1.40 |
| *Eunotia ambivalens* Lange-Bertalot & Tagliaventi | 1 | | | 0 | 2 | B | - | acf | - | i | - | ot | - | - | o | 1.00 |
| *Eunotia arcus* Ehrenberg | 1 | | | 0 | 2 | B | st-str | acf | 5.8–6.95 | i | temp | ot | ats | - | x-o | 0.50 |
| *Eunotia bilunaris* (Ehrenberg) Schaarschmidt | 1 | | | 0 | 2 | B | st-str | acf | 5.0–7.8 | i | temp | o-e | ate | es | o | 1.00 |
| *Eunotia diodon* Ehrenberg | 1 | | | 0 | 2 | B | st-str | acf | 6.75 | i | cool | ot | ats | - | x | 0.20 |
| *Eunotia exigua* (Brébisson ex Kützing) Rabenhorst | 1 | | | 0 | 2 | P-B, aer | st-str | acb | 3.4–8.0 | hb | temp | o-e | ate | es | x-o | 0.45 |
| *Eunotia hexaglyphis* Ehrenberg | 1 | | | 0 | 2 | B | - | acf | 5.79–6.66 | hb | temp | - | - | - | o-x | 0.70 |
| *Eunotia mucophila* (Lange-Bertalot, Nörpel-Schempp & Alles) Lange-Bertalot | 1 | | | 2 | 2 | P-B | st-str | acf | 5.25–6.4 | hb | temp | om | ate | - | o | 1.00 |
| *Eunotia praerupta* Ehrenberg | 1 | | | 2 | 0 | P-B | st-str | acf | 6.68–8.0 | hb | cool | om | ats | sx | x-o | 0.40 |
| *Fragilaria capucina* Desmazières | 1 | | | 2 | 2 | P-B | st-str | ind | 6.4–8.9 | i | temp | m | ats | es | b-o | 1.60 |
| *Frustulia crassinervia* (Brébisson ex W.Smith) Lange-Bertalot & Krammer | 1 | | | 0 | 2 | B | str | acf | 4.7–7.2 | hb | - | ot | ats | es | x-o | 0.50 |
| *Gomphonella calcarea* (Cleve) R.Jahn & N.Abarca | 1 | | | 2 | 0 | B | st-str | alf | - | i | - | om | ate | | b | 2.30 |
| *Gomphonella olivacea* (Hornemann) Rabenhorst | 1 | | | 2 | 0 | B | st-str | alf | 6.5–8.8 | i | temp | e | ate | es | o-b | 1.45 |
| *Gomphonema acuminatum* Ehrenberg | 1 | | | 0 | 2 | B | st-str | ind | 6.3–9.5 | i | temp | om | ats | es | o-b | 1.40 |
| *Gomphonema hebridense* W.Gregory | 1 | | | 0 | 2 | B | - | acf | 6.1 | - | - | - | - | - | - | - |
| *Gomphonema montanum* (Schumann) Grunow | 1 | | | 2 | 0 | B | str | ind | - | i | - | m | ats | es | x-b | 0.85 |
| *Gomphonema pala* E.Reichardt | 1 | | | 2 | 0 | B | - | - | - | - | - | - | - | - | o-b | 1.40 |
| *Gomphonema subclavatum* (Grunow) Grunow | 1 | | | 2 | 0 | B | str | ind | - | i | - | om | ats | es | o-b | 1.40 |
| *Gomphonema truncatum* Ehrenberg | 1 | | 3 | 0 | 2 | B | st-str | ind | 7.19 | i | temp | me | ats | es | o-b | 1.40 |
| *Gomphonema* sp. | 1 | | | 2 | 0 | - | - | - | - | - | - | - | - | - | - | - |
| *Gyrosigma attenuatum* (Kützing) Rabenhorst | 1 | | | 2 | 0 | P-B | st-str | alf | 6.9–8.5 | i | temp | om | ate | - | o-a | 1.80 |
| *Hannaea arcus* (Ehrenberg) R.M.Patrick | 1 | 2 | | 3 | 0 | B | str | alf | 5.7–7.5 | i | temp | om | ats | es | x | 0.30 |
| *Iconella linearis* (W.Smith) Ruck & Nakov | 1 | | | 2 | 2 | P-B | st-str | ind | 4.6–9.0 | i | - | om | ats | es | x-o | 0.50 |
| *Melosira undulata* (Ehrenberg) Kützing | 1 | | | 0 | 2 | P-B | - | ind | - | i | - | me | - | es | b | 2.00 |
| *Meridion circulare* (Greville) C.Agardh | 1 | | | 2 | 0 | P-B | st-str | ind | 6.6–8.3. | i | temp | om | ate | es | o | 1.10 |
| *Navicula cryptocephala* Kützing | 1 | | | 2 | 2 | P-B | st-str | ind | 6.5–8.4 | i | temp | o-e | ate | es | b | 2.10 |
| *Navicula cryptotenella* Lange-Bertalot | 1 | | | 2 | 0 | P-B | st-str | ind | 6.5–8.7 | i | temp | m | ats | es | o | 1.30 |
| *Navicula minima* Grunow | 1 | | | 2 | 2 | P-B | st-str | alf | 6.7–7.8 | hl | temp | e | hne | es | a-o | 2.60 |
| *Navicula pseudosilicula* Hustedt | 1 | | | 2 | 0 | P-B | - | ind | - | i | - | ot | - | - | o | 1.00 |
| *Navicula radiosa* Kützing | 1 | | | 2 | 2 | B | st-str | ind | 5–9 | i | temp | me | ate | es | b | 1.30 |
| *Neidium ampliatum* (Ehrenberg) Krammer | 1 | | | 0 | 2 | B | - | acf | 8.5–10.5 | hb | temp | ot | - | - | o | - |
| *Neidium bisulcatum* var. *subampliatum* Krammer | 1 | | | 2 | 0 | B | - | acf | – | hb | - | - | - | - | - | - |
| *Neidium dubium* (Ehrenberg) Cleve | 1 | | | 2 | 0 | B | str | alf | – | i | - | me | ats | - | b-o | 1.70 |
| *Nitzschia fonticola* (Grunow) Grunow | 1 | | | 2 | 0 | P-B | st-str | alf | 6.0–8.9 | i | temp | me | ate | - | o-b | 1.50 |
| *Nitzschia sublinearis* Hustedt | 1 | | | 2 | 0 | P-B | - | alf | – | i | - | me | hne | es | a | 3.00 |
| *Odontidium hyemale* (Roth) Kützing | 1 | | 1 | 0 | 2 | P-B | st-str | ind | 6.5–7.5 | hb | cool | ot | ats | sx | x | 0.30 |
| *Odontidium mesodon* (Kützing) Kützing | 1 | | | 2 | 0 | B | st-str | ind | 6.6–8.3 | hb | cool | ot | ats | sx | x-o | 0.40 |
| *Orthoseira dendroteres* (Ehrenberg) Genkal & Kulikovskiy | 1 | | | 2 | 0 | B, aer | - | - | – | i | - | - | - | - | x-o | 0.50 |

**Table 2.** *Cont.*

| Taxa | Epipelic | Epilithic | Epiphytic | AGL | KGL | Hab | Oxy | pH | pH Range | Sal | T | Tro | Aut-Het | D | Sap | S |
|---|---|---|---|---|---|---|---|---|---|---|---|---|---|---|---|---|
| *Pinnularia borealis* Ehrenberg | 1 | | | 2 | 2 | B, aer | st-str, aer | ind | 7.8 | i | | om | ate | es | x-o | 0.40 |
| *Pinnularia brebissonii* (Kützing) Rabenhorst | 1 | | | 0 | 2 | B | st-str | ind | - | i | temp | - | ats | - | p-a | - |
| *Pinnularia interrupta* W.Smith | 1 | | | 2 | 2 | B | st-str | ind | 6.0–8.0 | i | - | om | ats | - | o | - |
| *Pinnularia major* (Kützing) Rabenhorst | 1 | 1 | 1 | 2 | 2 | B | st-str | ind | 6.38–7.1 | i | temp | me | ate | | o-x | 0.60 |
| *Pinnularia microstauron* var. *nonfasciata* Krammer | 1 | | | 0 | 2 | B | - | - | - | - | - | - | - | - | - | - |
| *Pinnularia viridis* (Nitzsch) Ehrenberg | 1 | | | 2 | 0 | P-B | st-str | ind | 5.24–7.1 | i | temp | o-e | ate | es | x | 0.30 |
| *Stauroneis anceps* Ehrenberg | 1 | | 3 | 2 | 2 | P-B | st-str | ind | 4.8–8.0 | i | - | om | ate | sx | o | 1.30 |
| *Staurosira construens* Ehrenberg | 1 | | | 2 | 0 | P-B | st-str | alf | 5.5–9.0 | i | temp | me | ate | sx | o | 1.30 |
| *Staurosirella pinnata* (Ehrenberg) D.M.Williams & Round | 1 | | | 2 | 0 | P-B | st-str | alf | 6.2–9.3 | hl | temp | o-e | ate | es | o | 1.20 |
| *Surirella angusta* Kützing | 1 | | | 2 | 0 | P-B | st-str | alf | 6.9–8.9 | i | temp | e | ate | es | b-o | 1.70 |
| *Surirella robusta* Ehrenberg | 1 | | | 2 | 0 | P-B | st-str | ind | 7.6–9.5 | i | temp | ot | ats | es | o | 1.20 |
| *Tabellaria fenestrata* (Lyngbye) Kützing | 1 | | 3 | 0 | 2 | P-B | st-str | ind | 6.2 | i | - | om | ats | es | x | 0.30 |
| *Tabellaria flocculosa* (Roth) Kützing | 1 | | 3 | 1 | 1 | P-B | st-str | acf | 4.5–8.0 | i | eterm | ot | ats | es | o-x | 0.60 |

Note: Abbreviations: *: New record. Ecological preferences: Water temperature (T): cool, cool-loving species; temp, temperate temperature water inhabitants; eterm, eurythermic species; warm, warm water inhabitants. Habitat (Hab): B, benthic; P-B, planktonic–benthic; P, planktonic. Water pH (pH): acf, acidophilic species; ind, indifferent; alf, alkaliphilic species; acb, acidobiontes. Organic pollution, Watanabe diatom indicator system (D): sx, saproxenes, es, eurysaprobes; sp, saprophiles. Self-purification zone indicators (Sap): x/0.0—xenosaprobe; x-o/0.4—xeno-oligosaprobe; o-x/0.6—oligo-xenosaprobe; o/1.0—oligosaprobe; o-b/1.4—oligo-betamesosaprobe; x-b/0.8—xeno-betamesosaprobe; b-o/1.6—beta-oligosaprobe; o-a/1.8—oligo-alphamesosaprobe; b/2.0—betamesosaprobe; a-o/2.6—alpha-oligosaprobe; a/3.0—alphamesosaprobe; p-a/4.0—poly-alphamesosaprobe. Species–specific index of saprobity S (S). Trophic state: ot, oligotrafentic; o-m, oligo-mesotraphentic; m, mesotraphentic; me, meso-eutraphentic; e, eutraphentic; o-e, oligo- to eutraphentic. Nutrition type as Nitrogen uptake metabolism: ats, nitrogen-autotrophic taxa, tolerating very small concentrations of organically bound nitrogen; ate, nitrogen-autotrophic taxa, tolerating elevated concentrations of organically bound nitrogen; hne, facultative nitrogen-heterotrophic taxa, needing periodically elevated concentrations of organically bound nitrogen. Oxygenation (Oxy): str, streaming well oxygenated waters inhabitant; st-str, low streaming medium oxygenated waters inhabitant; st, standing low oxygenated water inhabitant. Water salinity (Sal): hb, halophobe; i, oligohalobious-indifferent; hl, oligohalobious-halophilous).

Only 15 diatom species were common to both lakes. Total floristic similarity between the neighbouring lakes was only 21.12%. Genera with the highest number of species were *Eunotia* (8), *Gomphonema* (6), *Pinnularia* (6), *Navicula* (5), and *Aulacoseira* (4). Other genera were represented by three or one species (Table 2, Figures 4 and 5).

The genus *Eunotia* (Figure 4) represented 11.26% of the diatom flora with eight species. *Eunotia mucophila* and *E. praerupta* species were common in both lakes, while other *Eunotia* species were identified only in Koçdüzü Great Lake (Table 1). This is a remarkable result because the water of the Koçdüzü Great Lake is alkaline (pH 8.45) (Table 1).

*Hannaea arcus* (Figure 4) was the only "common" determined species. For this species, which is generally considered to be oligosaprobic, with a wide range of pH and calcium bicarbonate content. This occurrance was rare. The diatoms, which were identified as single, constituted 97.18% of the flora.

The diatom assemblages of the Avusor Great Lake and the Koçdüzü Great Lake formed by cosmopolitan species. A major part of them is alpine and subalpine origin. This result is consistent with other studied high mountain lakes in the region. The diatom flora of both lakes is composed of indifferent (45.31%), alkaliphiles (31.25%), acidophiles (21.87%), and acidobiontes (1.56%) species. North alpine and alpine diatom species were also identified.

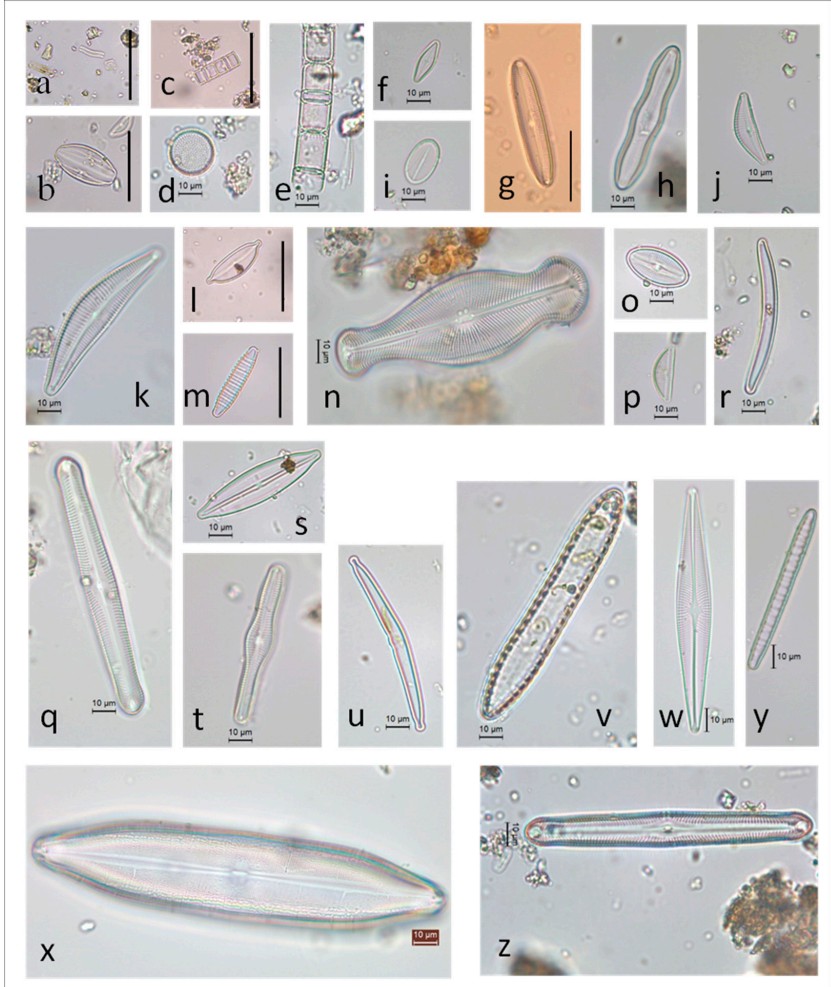

**Figure 4.** (**a**) *Achnanthidium minutissimum*, (**b**) *Amphora ovalis*, (**c**) *Aulacoseira italica*, (**d**) *A. lacustris*, (**e**) *A. valida*, (**f**) *Brachysira brebissonii*, (**g**) *Caloneis alpestris*, (**h**) *C. silicula*, (**i**) *Cocconeis placentula* var. euglypta, (**j**) *Cymbella affinis*, (**k**) *C. cistula*, (**l**) *Cymbopleura amphicephala*, (**m**) *Diatoma vulgaris*, (**n**) *Didymosphenia geminata*, (**o**) *Diploneis elliptica*, (**p**) *Encyonema minutum*, (**q**) *Epithemia gibba*, (**r**) *Eunotia mucophila*, (**s**) *Frustulia crassinervia*, (**t**) *Gomphonema montanum*, (**u**) *Hannaea arcus*, (**v**) *Iconella linearis*, (**w**) *Navicula radiosa*, (**x**) *Neidium dubium*, (**y**) *Odontidium hyemale*, and (**z**) *Pinnularia major*. Scale bar: 10 μm.

*3.3. Bioindicators Analysis*

Diatoms usually represent a major part of the algae community and all of them can be indicators of their environment. As can be seen in Table 2, the ecological preferences of each species in both lakes were revealed in respect to nine environmental variables. The bioindicator analysis was summarized for each lake in respect to each indicative variable. Figure 6 represents the distribution of indicator species number over the ecological groups in both studied lakes. The number of indicator species in Lake Avusor is always higher than in Koçdüzü Great Lake. Despite this, the overall distribution in each group of indicators by the displayed parameter shows the most representative groups when a trend line is plotted. That is, we see that among the indicators of confinement to a particular substrate, the largest number of species are benthic or planktonic–benthic in both lakes. The water in both lakes is well saturated with oxygen. However, the pH of the water in Avusor Great Lake is noticeably higher than in Koçdüzü Great Lake, as seen in the trend lines. Both lakes are fresh and of moderate temperature. Two separate groups represent the indicators of the trophic state in both lakes with a predominance of oligo-mesotrophic species in one and meso-eutrophic species in the other. Diatom communities in both lakes are represented by

autotrophic species. In both lakes, organic pollution according to Watanabe indicators is characterized as low or moderate, and in Avusor Great Lake, there are more indicators of xenosaprobes, which indicates clearer waters. Indicators of organic pollution according to V. Sládeček [47] belonged to four classes of water quality with a predominance of a group of indicators of class 2 in both lakes. Moreover, there were twice as many of class 2 indicators in Avusor Great Lake than in Koçdüzü Great Lake, which indicates cleaner waters in the first lake. At the same time, the saprobity index C calculated by us for the abundance of indicator species in the community of each of the lakes and the species–specific index is 1.08 in Avusor and 0.87 in Koçdüzü. This shows that self-purification was more active in Avusor compared to Koçdüzü, but at the same time, it characterizes the waters of both lakes as clean, since the index values differ slightly and belong to the 2nd class of water quality [48].

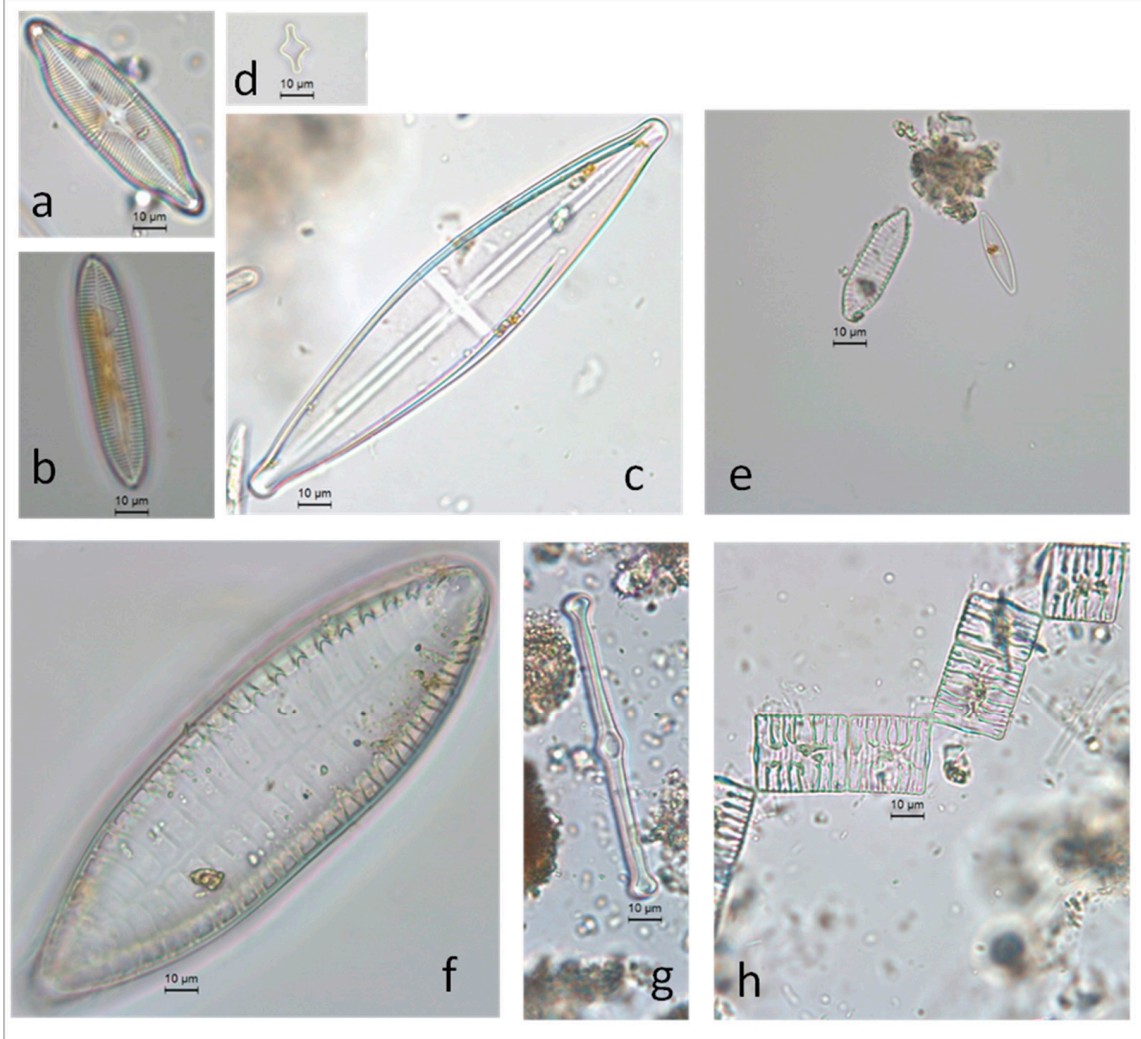

**Figure 5.** (**a**) *Pinnularia microstauron* var. nonfasciata, (**b**) *P. viridis*, (**c**) *Stauroneis anceps*, (**d**) *Staurosira construens*, (**e**) *Surirella angusta*, (**f**) *S. robusta*, (**g**) *Tabellaria fenestrata*, and (**h**) *T. flocculosa*. Scale bar: 10 μm.

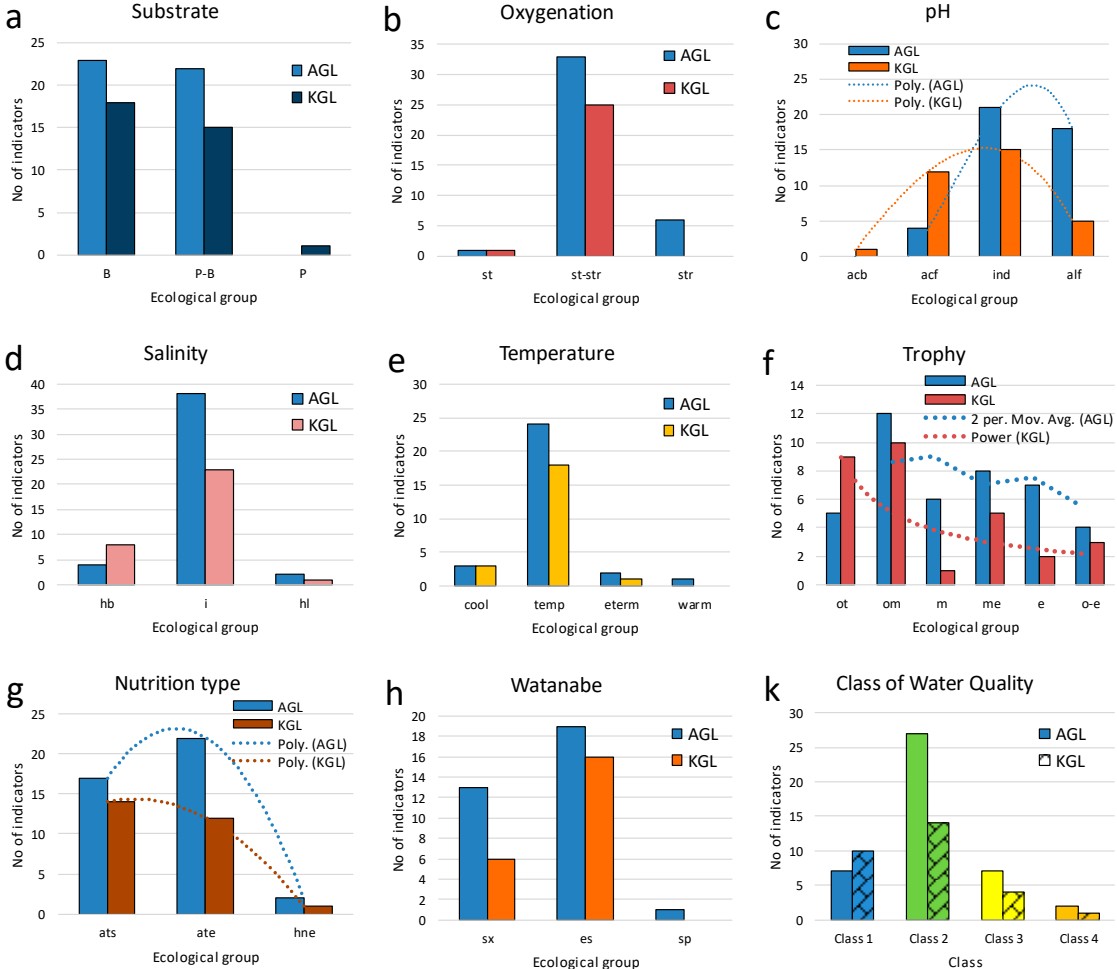

**Figure 6.** Distribution of indicator species number over the ecological groups in the Avusor Great Lake (AGL) and Koçdüzü Great Lake (KGL). Abbreviations: Substrate (**a**) B, benthic; P-B, planktonic–benthic; P, planktonic. Oxygenation (**b**) str, streaming well oxygenated waters inhabitant; st-str, low streaming medium oxygenated waters inhabitant; st, standing low oxygenated water inhabitant. Water pH (**c**) acf, acidophilic species; ind, indifferent; alf, alkaliphilic species; acb, acidobiontes. Water salinity (**d**) hb, halophobe; i, oligohalobious-indifferent; hl, oligohalobious-halophilous). Water temperature (**e**) cool, cool-loving species; temp, temperate temperature water inhabitants; eterm, eurythermic species, warm, warm water inhabitants. Trophic state (**f**) ot, oligotrafentic; o-m, oligo-mesotraphentic; m, mesotraphentic; me, meso-eutraphentic; e, eutraphentic; o-e, oligo- to eutraphentic. Nutrition type as Nitrogen uptake metabolism (**g**) ats, nitrogen-autotrophic taxa, tolerating very small concentrations of organically bound nitrogen; ate, nitrogen-autotrophic taxa, tolerating elevated concentrations of organically bound nitrogen; hne, facultative nitrogen-heterotrophic taxa, needing periodically elevated concentrations of organically bound nitrogen. Organic pollution, Watanabe diatom indicators (**h**) sx, saproxenes, es, eurysaprobes; sp, saprophiles. Class of water quality based on species–specific saprobity index S (**k**) Class 1, S = 0.0–0.5; Class 2, S = 0.5–1.5; Class 3, S = 1.5–2.5; Class 4, S = 2.5–3.5.

Figure 7 is a generalized view of the bioindicative characteristics of both lakes. The heat map was based on the distribution of the abundance of indicator species by ecological groups. The abundance of indicator species in Avusor Great Lake is generally higher. The predominance of indifferent indicators in terms of salinity and moderately oxygenated waters is well expressed (red). At the same time, the abundance of a group of plankton–benthic and benthic species, as well as indifferences in water pH and the second quality

class (yellow) is confirmed by an analysis of the distribution of the number of indicator species.

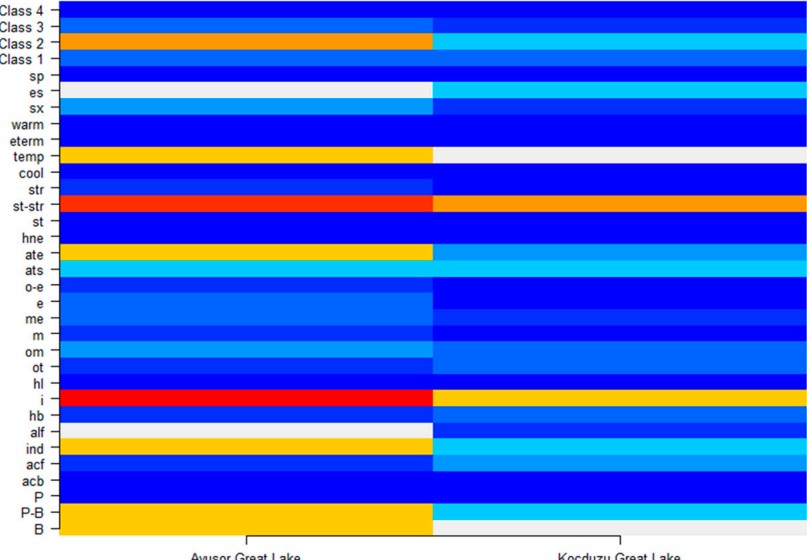

**Figure 7.** Comparative heat map of indicator species distribution in the ecological groups of the Avusor Great Lake (AGL) and Koçdüzü Great Lake (KGL). The heatmap temperature ranges from 0 (white) to 33 (red). Abbreviations from down to up on axis y: Substrate: B, benthic; P-B, planktonic–benthic; P, planktonic. Water pH: acf, acidophilic species; ind, indifferent; alf, alkaliphilic species; acb, acidobiontes. Water salinity: hb, halophobe; i, oligohalobious-indifferent; hl, oligohalobious-halophilous). Trophic state: ot, oligotrafentic; o-m, oligo-mesotraphentic; m, mesotraphentic; me, meso-eutraphentic; e, eutraphentic; o-e, oligo- to eutraphentic. Nutrition type as Nitrogen uptake metabolism: ats, nitrogen-autotrophic taxa, tolerating very small concentrations of organically bound nitrogen; ate, nitrogen-autotrophic taxa, tolerating elevated concentrations of organically bound nitrogen; hne, facultative nitrogen-heterotrophic taxa, needing periodically elevated concentrations of organically bound nitrogen. Oxygenation: str, streaming well oxygenated waters inhabitant; st-str, low streaming medium oxygenated waters inhabitant; st, standing low oxygenated water inhabitant. Water temperature: cool, cool-loving species; temp, temperate temperature water inhabitants; eterm, eurythermic species, warm, warm water inhabitants. Organic pollution, Watanabe: sx, saproxenes, es, eurysaprobes; sp, saprophiles. Class of water quality based on species–specific index saprobity S: Class 1, S = 0.0–0.5; Class 2, S = 0.5–1.5; Class 3, S = 1.5–2.5; Class 4, S = 2.5–3.5.

### 3.4. Comparative Floristic Analysis

In order to identify the features of the studied flora of two glacial high mountain lakes, a comparative floristic analysis was carried out using statistical methods. For calculations, floras of only diatoms were selected from similar habitats, that is, lakes located in the same climatic region. A general list of diatom floras was compiled for (1) the studied lakes, (2) the high mountain lakes of the Artabel Nature Park [42], and (3) some lakes of central and northern Turkey [49]. All species in the list have been brought into line with modern taxonomy [32]. Then, the Bray–Curtis similarity indices were calculated, and a similarity tree was built, and then a correlation analysis was carried out in the JASP 0.16.4 program.

Figure 8 demonstrate the similarity level of compared diversity of diatoms in the lakes of Turkey. Two of the clusters divided studied lists of diatoms into the group of other lakes in northern Turkey (cluster 1) placed on the altitude 30–1512 m a.s.l. and group of high mountain lakes on altitude 2382–2678 m a.s.l. (cluster 2). The flora of the studied lakes, Avusor and Koçdüzü, was included in cluster 2.

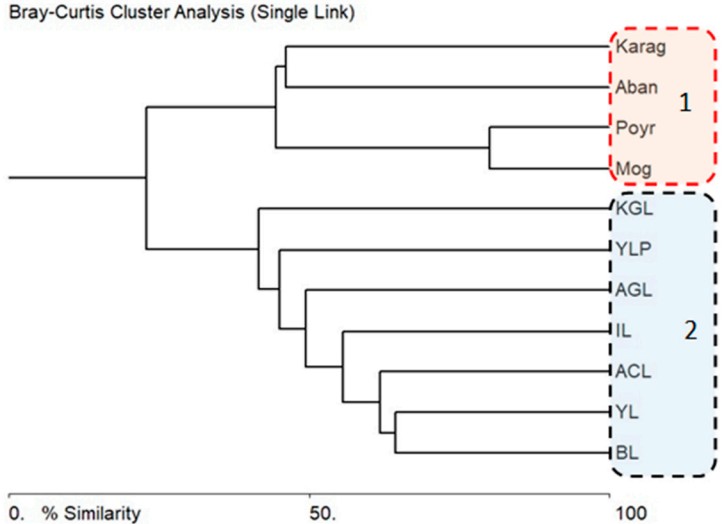

**Figure 8.** Bray-Curtis similarity analysis of species content in the Artabel Lakes in Gümüşhane province (Artabel Lakes (ARL), Beş Lakes (BL), Yıldız Lakes (YL), Acembol Lakes (ACL), İsimsiz Lake (IL), and Yıldız Lake (YLP)) according [42], the Avusor Great Lake (AGL), Koçdüzü Great Lake (KGL), and other lakes in northern Turkey (Mogan (Mog), Abant (Aban), Poyrazlar (Poyr), Karagol (Karag)) according [49].

A JASP Network plot of species composition correlation of studied lakes AGL and KGL in Rize with other alpine lakes in Gümüşhane province and other lakes in northern Turkey (Figure 9) shows two different clusters. Cluster 1 combined other lakes in northern Turkey diatom floras and cluster 2 included the mountain floras group to which the diatoms of Avusor and Koçdüzü lakes are also included.

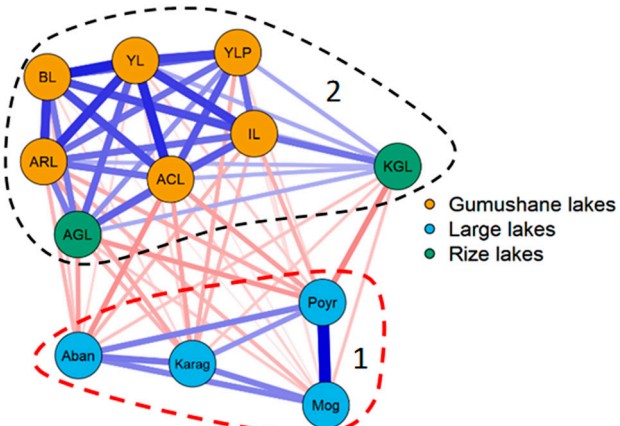

**Figure 9.** JASP Network plot of correlation on the level more than 50% for species composition of the lakes in Rize province, of the Artabel Lakes in Gümüşhane province, and other lakes in northern Turkey. Abbreviations: Artabel Lakes (ARL), Beş Lakes (BL), Yıldız Lakes (YL), Acembol Lakes (ACL), İsimsiz Lake (IL), and Yıldız Lake (YLP) according [42]; the Avusor Great Lake (AGL), Koçdüzü Great Lake (KGL); other lakes in northern Turkey: Mogan (Mog), Abant (Aban), Poyrazlar (Poyr), Karagol (Karag) according [49]. The line thickness between lakes reflects the correlation value (represented significant only); blue is positive, red is negative. Clusters are outlined by dashed lines.

The data on the species content in the studied lakes and some other diatom floras of the similar climatic zone lakes were combined in Table 3 to reveal the role of altitude in the formation of diatom floras. All combined lakes are protected as a part of the natural parks. It shows that species richness of diatoms was similar in the presented lakes. At the same time, the lake altitude ranged between 30 and 2980 m a.s.l. All other available variable

values fluctuated in a narrow range. Calculation of Pearson correlation coefficients [39] revealed only one variable—electrical conductivity of the lake water, which was negatively correlated to the lake altitude (r = −0.975; $p < 6.87 \times 10^{-8}$). All other variables have no significant correlation.

**Table 3.** Distribution of major environmental variables (averaged) and the diatom species number in the studied lakes Avusor and Koçdüzü, Artabel Lakes and other lakes in northern Turkey for comparative analysis.

| Lake | Altitude, m a.s.l. | Temperature, °C | DO, mg L$^{-1}$ | pH | Conductivity, µSm cm$^{-1}$ | No Species |
|---|---|---|---|---|---|---|
| Poyrazlar | 30 | 14.6 | 6.98 | 7.90 | 425 | 44 |
| Mogan | 972 | 14.3 | 11.50 | 8.80 | 258 | 41 |
| Abant | 1350 | 16.5 | 8.80 | 8.25 | 264 | 41 |
| Karagol | 1512 | 18.6 | 6.90 | 8.01 | 138.5 | 76 |
| KGL | 2382 | 21.0 | 9.20 | 8.45 | 104.7 | 34 |
| IL | 2668 | 19.1 | 4.25 | 6.78 | 12.0 | 38 |
| AGL | 2678 | 15.9 | 10.2 | 7.58 | 45.3 | 46 |
| ACL | 2712 | 16.2 | 2.80 | 7.20 | 32.3 | 43 |
| ARL | 2834 | 15.3 | 6.60 | 7.00 | 33.4 | 53 |
| BL | 2883 | 13.0 | 9.00 | 7.00 | 26.0 | 46 |
| YLP | 2980 | 14.5 | 2.34 | 7.20 | 29.2 | 21 |
| YL | 2980 | 13.4 | 2.80 | 6.90 | 26.6 | 41 |

Note. Abbreviations: Artabel Lakes (ARL), Beş Lakes (BL), Yıldız Lakes (YL), Acembol Lakes (ACL), İsimsiz Lake (IL), and Yıldız Lake (YLP) according [42]; the Avusor Great Lake (AGL), Koçdüzü Great Lake (KGL); other lakes in northern Turkey: Mogan (Mog), Abant (Aban), Poyrazlar (Poyr), Karagol (Karag) according [49].

In this case, the surface plots were constructed to follow the analysis. Figure 10 show that diatom species richness can be greater if the altitude decreased but the water temperature increased (Figure 10a) and if water pH decreased (Figure 10b). Species number also increased in lower altitude and dissolved oxygen (Figure 10c) and lower conductivity (Figure 10d).

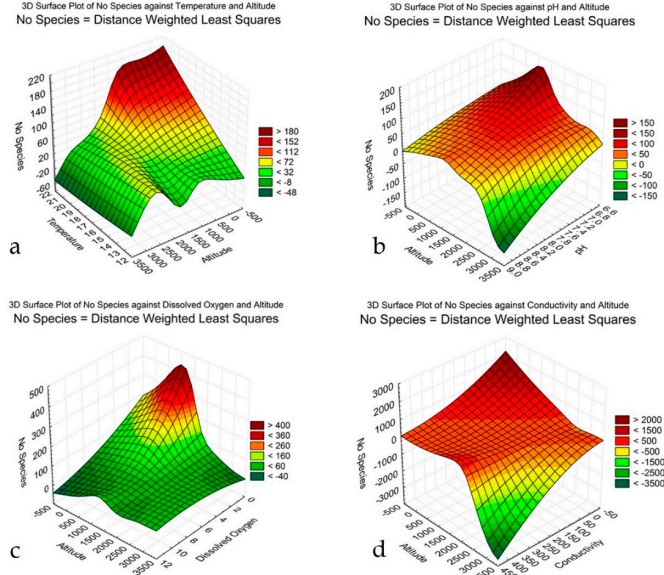

**Figure 10.** Distribution of diatom species richness over major environmental variables in the lakes of Rize province, the Artabel Lakes in Gümüşhane province, and other lakes in northern Turkey based on Table 3. Distance Weighted Least Squares dependence of species richness on environmental parameters: (**a**), water temperature and altitude; (**b**), pH and altitude; (**c**), Dissolved oxygen and altitude; (**d**), water conductivity and altitude.

At the same time, the interrelations in environmental variables were revealed and represented in Figure 11. Dissolved oxygen as one of the important factors for diatoms development, was slightly increased with the water temperature decreasing (Figure 11a) but with its distribution over the lake altitude having a threshold in altitude 2000 m, after which oxygen concentration can rapidly decrease. Lake water temperature was an indifferent factor for electrical conductivity that decreased with altitude increasing (Figure 11b).

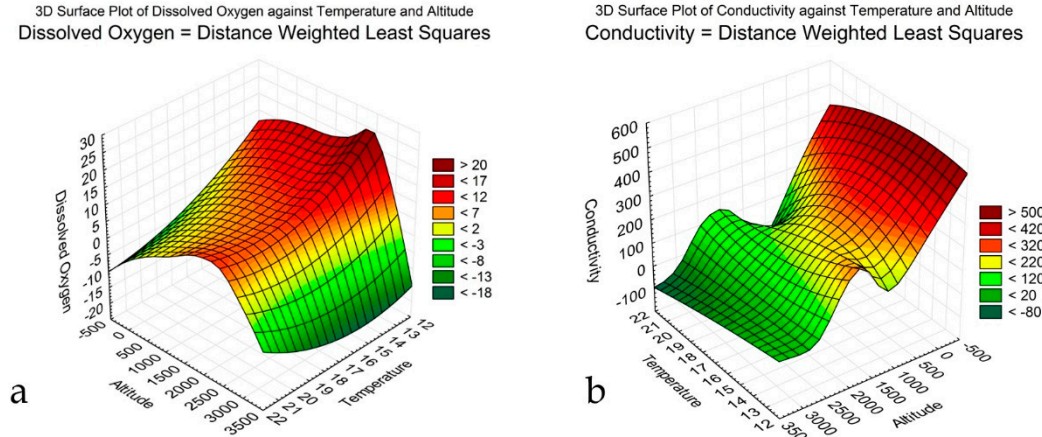

**Figure 11.** Distribution of major environmental variables over altitude of the lakes in Rize province, the Artabel Lakes in Gümüşhane province, and other lakes in northern Turkey based on Table 3. Distance Weighted Least Squares dependence of environmental parameters: (**a**), dissolved oxygen and water temperature and altitude; (**b**), water conductivity and temperature and altitude.

## 4. Discussion

The influence of climatic gradients on the freshwater algae community has been best studied for diatoms [50], which usually contain more indicator species than other aquatic organisms [51]. The recognition of the influences and major impacted factors now can be used with many different approaches, the majority of which are statistics and bioindication [52].

The objects of our research were diatoms of benthos and periphyton of glacial lakes at an altitude of about 3000 m in the mountains of northern Turkey. The identified diversity, including 71 taxa of species rank and below, turned out to be similar in species richness to other high mountain lakes, such as Artabel, as well as some others in northern Turkey [42,49]. The head part of studied diatom flora of both lakes represented only four genera species: *Eunotia* (8), *Gomphonema* (6), *Pinnularia* (6), and *Navicula* (5) that is similar to the other high mountain lakes in Turkey and Europe [4,42,53–59]. With the dominance of *Eunotia* species, as well as the noticeable presence of *Pinnularia*, which both prefer low conductivity and slightly acidic freshwater [60], the results of this study characterize the lake water as mostly circumneutral and acidic and have low mineral content [38].

The dominating species of *Eunotia* have been important in diatom flora and also in other high mountain lakes in the Eastern Black Sea region [42,55,56] because they have favorable conditions in which *Eunotia* species can grow. The members of this genus prefer waters with acidic or circumneutral pH and low or medium conductivity [29,38,61,62]. Water pH and conductivity values of water are the main factors in the distribution of individuals of this genus; water temperature is not effective [63]. We think that this situation is because the *Eunotia* species reached the lake by streams and adapted successfully to this pH value. In addition, *Eunotia hexaglyphis* has been identified as a new record for the Freshwater Algal Flora of Turkey. Detailed knowledge about this species were given in a separate paper [64]. As has been revealed for studied lakes in Rize, the dominated species of *Eunotia* and *Pinnularia* prefer low conductivity and slightly acidic fresh waters while the genus *Gomphonema* occurs in circumneutral lakes and streams [37,38].

*Hannaea arcus*, which was abundant in both studied lakes, does not prefer acidic freshwater habitats and tolerates some organic enrichment [31]. It was found in the epilithic samples of the Artabel Great Lake. *Encyonema minutum* (Figure 4), which was found in the epipelic and epilithic habitats of the Avusor Great Lake, has a wide geographic distribution [25] and prefers circumneutral waters [65]; this was rare. *Tabellaria flocculosa* (Figure 5) was recorded in both lakes. This species, which has a wide tolerance, is found in acidic, oligotrophic, and mesotrophic waters [25]. According to H. Van Dam et al. [65], it occurs in acidophilic, β-mesosaprobic, and mesotrophic waters. Therefore, the dominating species can characterize the lake water as having low conductivity and slightly acidic freshwaters.

Regarding eutrophication and the effects of acidification that are among the important problems faced by glacial lakes [66,67], these problems were not identified in either lake based on the bioindication and chemical data results. The statistical heat map of indicator species abundance shows the similarity of the results with the analysis of the distribution of the number of species, which indicates the low diversity of the identified diatoms, on the one hand, and the absence of a pronounced domination in the communities, on the other hand [68]. This is most often inherent in undisturbed ecosystems of natural clean lakes, especially high mountains [8,9,69].

The critical altitude for species richness in the Hindukush piedmonts of Pakistan was about 1400 m a.s.l. [70]. However, in the Caucasus Mountains, this barrier was found at an altitude of about 2000 m [44]. Our investigation results confirm that diatom diversity in northern Turkey lakes increased in species richness before 2000 m a.s.l. and decreased after this altitude.

The diatoms of the studied glacial lakes in the province of Rize form a separate group of high mountain lakes in northern Turkey in terms of taxonomic composition and bioindicators. Consequently, the environmental conditions of high mountains act as selection factors on the flora of diatoms, which was noted earlier [8,9,44,69,71].

Comparison of the influence of individual environmental parameters on diatom communities in the 12 lakes showed that the northern Turkey environment influenced diatoms that prefer waters of temperate temperature, circumneutral pH, moderately saturated with oxygen, and medium content of mineral ions. At the same time, the altitude negatively affects the species richness of diatoms as was revealed in Pamir lakes [9] and 2000 m was found as an environmental barrier upper of which diatom communities in mountain lakes partly lost their diversity as in Georgia [44,45,50].

## 5. Conclusions

An analysis of the diatom community using statistical and bioindicator methods has identified water salinity and lake altitude as regulatory factors that can negatively affect diatom diversity in Turkey's high mountain lakes. Thus, we only partially confirmed our hypothesis after analyzing the composition of the diatom community. The diatom flora of the high mountain lakes has its own peculiarities, different from the lowland lakes of northern Turkey. Increasing water purity and decreasing diatoms diversity in high mountain lakes will mean that the lakes can be safe and productive sources of water in the face of future warming.

**Author Contributions:** Conceptualization, B.Ş. and S.B.; methodology, B.Ş. and S.B.; software, S.B.; validation, B.Ş. and S.B.; formal analysis, B.Ş. and S.B.; investigation, B.Ş.; resources, B.Ş.; data curation, B.Ş.; writing—original draft preparation, B.Ş. and S.B.; writing—review and editing, B.Ş. and S.B.; visualization, B.Ş. and S.B.; supervision, B.Ş.; project administration, B.Ş.; funding acquisition, B.Ş. All authors have read and agreed to the published version of the manuscript.

**Funding:** This research received no external funding.

**Data Availability Statement:** The data presented in this study are available on request from the authors.

**Acknowledgments:** This work was partly supported by the Israeli Ministry of Aliyah and Integration.

**Conflicts of Interest:** The authors declare no conflict of interest.

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
