# Peer review of "Role of Altitude in Formation of Diatom Diversity of High Mountain Protected Glacier Lakes in the Kaçkar Mountains National Park, Rize, Turkey"

_environments, doi:10.3390/environments9100127_

Round 1
Reviewer 1 Report
I would like to suggest only one correction required. The names of authors of all listed species should be checked and improved (abbreviated in appropriate way) according IPNI https://www.ipni.org I recommend accept this paper after improvement of this issue.Author Response
Dear Editor and the Reviewer 1,
Thank you for your recommendations.
Please find below the response to comments.
With best regards,
Prof Sophia Barinova,
Corresponding author
Response to Reviewer 1:
Dear Dr. Barinova,
Your manuscript has been reviewed by experts in the field and we request that you make major revisions before it is processed further. Please revise your manuscript according to the reviewers' comments and upload the revised file within 10 days. Please click on the "Peer Review Reports" below to find the reviewers' comments and the version of your manuscript to be used for your revisions.
I would like to suggest only one correction required. The names of authors of all listed species should be checked and improved (abbreviated in appropriate way) according IPNI https://www.ipni.org I recommend accept this paper after improvement of this issue.
Response: Resource IPNI is most related to the Plant names and have not majority of diatom algae names. In contrary, as we mentioned in ms, “The current scientific names of the species were updated according to algaebase.org [32]”, which covered whole algae diversity.
Reviewer 2 Report
The article "The Role of altitude in formation of diatom diversity of high-mountain protected glacier lakes in the Kackar Mountains National Park, Rize, Turkey" is a paper that contains useful data about diatoms found in high altitude lakes in Turkey. The field and lab methods and literature used are appropriate. However, the current state of the manuscript has several flaws that must be addressed before it can be published.
Major Issues
1. None of the methods for calculating the various indicators are found in the methods section.
2. Discussion is far too short and it does not get to the heart of the question that the authors are asking, "how does altitude affect diatom community composition?"
3. 3-D plots have all kinds of negative numbers that are impossible.
4. Revised manuscript should be read by a native English speaker. There are a lot of awkward constructions and word choices throughout the manuscript that hampers the reader's understanding of the science. See marked up manuscript for some grammatical/ syntactical suggestions.
Minor Issues
1. See marked-up manuscript (attached)

Author Response
Dear Editor and the Reviewer 2,
Thank you for your recommendations.
Please find below the response to the comments.
With best regards,
Prof Sophia Barinova,
Corresponding author
Responses to Reviewer 2:
The article "The Role of altitude in formation of diatom diversity of high-mountain protected glacier lakes in the Kackar Mountains National Park, Rize, Turkey" is a paper that contains useful data about diatoms found in high altitude lakes in Turkey. The field and lab methods and literature used are appropriate. However, the current state of the manuscript has several flaws that must be addressed before it can be published.
Major Issues
- None of the methods for calculating the various indicators are found in the methods section.
Response: Added Distance Weighted Least Square method for 3D plots construction, Willis curve method, Ecological preferences method extended.
- Discussion is far too short and it does not get to the heart of the question that the authors are asking, "how does altitude affect diatom community composition?"
Response: The discussion includes links to information about the relationship between the dynamics of algae diversity and the altitude of the habitat. There are not many such studies. That is why we considered it important to carry out this work with a focus on comparing the influence of environmental factors on the distribution of diatom diversity. In the discussion, references are given and our results are compared only with those works where there is data on the relationship between the diversity of diatom communities and the altitude of the habitat. Very few works have been carried out in high-mountain lakes due to their inaccessibility. Thus, in the discussion, concentrated comparisons of our materials with similar ones from other sources are given. You can expand the text volume, but this will not lead to other conclusions, but only increase the volume of ms.
- 3-D plots have all kinds of negative numbers that are impossible.
Response: Figures 10 and 11 are constructed based on the data for each of the parameters shown in Table 3. The construction is done in the Program Statistica 12.0 using the Distance Weighted Least Square method. For comparison, one main parameter and two others are selected for each graph, within the limits of which the program calculates probable changes in the Main parameter. Thus, the resulting graph shows the trends in each of the related parameters. From here, outrageous values ​​may appear, which are not real, but only reflect trends for a given distribution. The graph can be interpreted as a trend of change (increase or decrease) in the values ​​of the main parameter (on the x axis) when the other two parameters (y and z axes) change.
- Revised manuscript should be read by a native English speaker. There are a lot of awkward constructions and word choices throughout the manuscript that hampers the reader's understanding of the science. See marked up manuscript for some grammatical/ syntactical suggestions.
Response: English edited, text corrected.
Minor Issues
- See marked-up manuscript (attached)
Response: All comments was included to corrected ms.
Reviewer 3 Report
The matter is of interest but the presentation and discussion of results should be better presented; see detailed comments in attached file

Author Response
Dear Editor and the Reviewer 3,
Thank you for your recommendations.
Please find below the response to comments.
With best regards,
Prof Sophia Barinova,
Corresponding author
Response to Reviewer 3:
Comments and Suggestions for Authors:
The matter is of interest but the presentation and discussion of results should be better presented; see detailed comments in attached file
Response: Many additions and explanations were added to ms for better understanding.
Pag 1
row 21 autotrophic species: I am not expert of Diatoms,
Response: Diatoms are autotrophic algae with the mixotrophy possibility by some species. In any case, we delete it because this characteristic is not so significant in the investigation.
row 23 Index saprobity: what index ?
Response: Index saprobity S is the organic pollution index.
row 29 avoid to repeat keywords in words given in title (es glacier lakes)
Response: Corrected.
row 36 what do you mean as “global ecosystems” ?
Response: deleted
row 42 always oligotrophic ?
Response: Corrected as: usually.
Pag 2
Row 60 replace “in all seasons” with “during the whole year”
Response: Corrected.
Row 65-73 it should be supported the statement that altitude is related to oxygen saturation and ionic components, generally altitude is related to water temperature that influences oxygen concentration not oxygen saturation and ionic component is generally not necessarily related to altitude
Response: Corrected.
Row 75 delete “.”
Response: Deleted sentence.
Pag 3
Row 93: why only one lake, the other lake had no epilithic substrates ?
Response: Yes, second lake it have not because have no stones.
Pag 4
Row 114 what means “botnet package” ? is OK ? It is better to state that some components were below detection method instead of “absent”
Response: “botnet package” is OK as recommended in this program citation. Methods added to the MM part.
Pag 5
Row 144 in figure 3 legend add something as “continuous” = observed, “dotted” = expected
Response: Added.
Row 150 delete “studied”
Response: Deleted.
Row 158 “Watanabe” I do not know this word
Response: Added: Watanabe diatom indicator system.
Table 2: it is rather difficult to read, can be divided into two tables ?
Response: Table 2 is a concentrated material as a basis for subsequent calculations and constructions. It includes all identified species with frequency scores, substrate characteristics, and ecological preferences of each species. If it is divided by two, this will not increase the information content, but will significantly increase the volume. That is, the article will be difficult to read.
Pag 15
Fig 7: heatmap must be better specified: 0 white, ? blue, ? yellow, ? orange, 33 red , Labels on the left of figure are too small, cannot be read
Response: The Heatmap figure is enlarged in visual size. However, it is given in sufficient resolution so that, by increasing the size of the manuscript on the screen, it is easy to read all the designations.
The color scale is qualitative and is presented to compare the characteristics of the two lakes. It does not imply quantitative relationships between parameters and color, since it is difficult to characterize transitional colors.
Pag 18
Figure 10: these figures are very elegant but in my opinion do not aid so much in understanding the situation
Response: Figures 10 and 11 make it easy to follow the direction of parameter changes relative to each other. We constructed these figures as statistical trends of changes in diatom species richness in the lakes of the northern part of Turkey (Figure 10) relative to the main environmental parameters: oxygen, pH, conductivity, and temperature. Also, Figure 11 shows the statistical trends of interdependencies between environmental indicators, which is impossible to identify in Table 3.
The relation between environmental factors (altitude, water temperature, dissolved oxygen, pH nutrient etc) and species composition is surely complex and the results can be limited to the particular database selected; if you do not see a monotone relation between air or water temperature and altitude, it can be due to several reasons, for example to the season of sampling (generally high altitude sites are sampled in summer, low ones in spring.
Response: Of course, the relationships are complex and can only be traced by building trends in statistical programs. However, for high-mountain lakes, as you rightly noted, the sampling season falls on the summer, the vegetation peak of community development. Thus, having no seasonal changes, but only summer, it can be said that, even at the peak of development, certain environmental factors are of the greatest importance.
My suggestion is to explain better how you have created the database used for the comparison
Response: We added detailed explanation of data base for comparison in the beginning of paragraph 3.4. Comparative floristic analysis.
Round 2
Reviewer 2 Report
The materials and methods section does not have the methods used to calculate each of the indices found in Figure 6 or even a citation for each. Please revise giving the methodology for determining each index with appropriate citation.
Author Response
Response 2 to the Reviewer 2
Dear Editor and the Reviewer 2,
Thank you very much for reviewing my ms. Please find below my response to your comment.
With best regards,
Prof Sophia Barinova,
Corresponding author
The materials and methods section does not have the methods used to calculate each of the indices found in Figure 6 or even a citation for each. Please revise giving the methodology for determining each index with appropriate citation.
Response: To build histograms, not indices were used, but the number of species with the same ecological properties in each group of indicators. The histograms are constructed on the basis of Table 2. In the MM section, the works are cited from which the data on the ecological properties of the species under the numbers [36-38] are taken.
Additional explanations are given in the MM section.
For each species, its indicator properties were determined in relation to one or more environmental variables. Then the data on the number of species in each indicator group were summarized. The distribution of the number of species with the same indicator properties was constructed with respect to each environmental variable. Class of water quality indicators were grouped on the range of species-specific index saprobity S: Class 1, S=0.0-0.5; Class 2, S=0.5-1.5; Class 3, S=1.5-2.5; Class 4, S=2.5-3.5.

Reviewer 3 Report
I am satisfied of your rebuttal letter and corrections, I can accept the manuscript
Author Response
Response to the Reviewer 3.
Dear Editor and the Reviewer 2,
Thank you very much for reviewing my ms. Please find below my response to your comment.
I am satisfied of your rebuttal letter and corrections, I can accept the manuscript
Response: Thank you very much for reviewing my ms.
With best regards,
Prof Sophia Barinova,
Corresponding author

Round 3
Reviewer 2 Report
The methods to calculate each of the indices shown in Figure 6 must be included. There should be an equation for their calculation in the materials and methods section.
Author Response
Response 3 to the Reviewer 2
Dear Editor and the Reviewer 2,
Thank you very much for reviewing my ms. Please find below my response to your comment.
With best regards,
Prof Sophia Barinova,
Corresponding author
Reviewer 2 question:
The methods to calculate each of the indices shown in Figure 6 must be included. There should be an equation for their calculation in the materials and methods section.
Response: Figure 6 represent the number of species in each group of indicators. This calculation was done as simple sum of indicators in each category that given in Table 2. So, it need not given the equation because it is not indices. As we mentioned in our previous response:
To build histograms, not indices were used, but the number of species with the same ecological properties in each group of indicators. The histograms are constructed based on Table 2. In the MM section, the works are cited from which the data on the ecological properties of the species under the numbers [36-38] are taken.
Additional explanations are given in the MM section:
For each species, its indicator properties were determined in relation to one or more environmental variables. Then the data on the number of species in each indicator group were summarized. The distribution of the number of species with the same indicator properties was constructed with respect to each environmental variable. Class of water quality indicators were grouped on the range of species-specific index saprobity S: Class 1, S=0.0-0.5; Class 2, S=0.5-1.5; Class 3, S=1.5-2.5; Class 4, S=2.5-3.5.
